# Shall We Pretrain Autoregressive Language Models with Retrieval? A Comprehensive Study

**Boxin Wang**[*‡1] **Wei Ping**[*†2] **Peng Xu**[*2] **Lawrence McAfee**[2]

**Zihan Liu**[2] **Mohammad Shoeybi**[2] **Yi Dong**[2] **Oleksii Kuchaiev**[2]

**Bo Li**[1] **Chaowei Xiao**[2,3] **Anima Anandkumar**[2] **Bryan Catanzaro**[2]

## Abstract

Large decoder-only language models (LMs) can be largely improved in terms of perplexity by retrieval (*e.g.*, RETRO), but its impact on text generation quality and downstream task accuracy is unclear. Thus, it is still an open question: *shall we pretrain large autoregressive LMs with retrieval?* To answer it, we perform a comprehensive study on a *scalable pretrained* retrieval-augmented LM (i.e., RETRO) compared with standard GPT and retrieval-augmented GPT incorporated at fine-tuning or inference stages. We first provide the recipe to reproduce RETRO up to 9.5B parameters while retrieving a text corpus with 330B tokens. Based on that, we have the following novel findings: *i)* RETRO outperforms GPT on text generation with much less degeneration (i.e., repetition), moderately higher factual accuracy, and slightly lower toxicity with a nontoxic retrieval database. *ii)* On the LM Evaluation Harness benchmark, RETRO largely outperforms GPT on knowledge-intensive tasks, but is on par with GPT on other tasks. Furthermore, we introduce a simple variant of the model, RETRO++, which largely improves open-domain QA results of original RETRO (e.g., EM score +8.6 on Natural Question) and significantly outperforms retrieval-augmented GPT in both fine-tuning and zero-shot evaluation settings. Our findings highlight the promising direction of pretraining autoregressive LMs with retrieval as future foundation models. We release our implementation at: `https://github.com/NVIDIA/Megatron-LM#retro`.

## 1 Introduction

Large language models (LMs), including masked LMs (e.g., BERT (Devlin et al., 2018)), autoregressive LMs (e.g., GPT (Brown et al., 2020)), and encoder-decoder LMs (e.g., T5 (Raffel et al., 2020), BART (Lewis et al., 2020a)), have obtained state-of-the-art results for various NLP tasks. Among them, the autoregressive LMs like GPT-3 (Brown et al., 2020) and GPT-4 (OpenAI, 2023) demonstrate noticeable in-context learning ability and excellent long-form text generation results. Due to its importance, the community has spent considerable efforts to scale up such autoregressive generative LMs with more data and parameters and observed significant breakthroughs in a variety of real-world applications (e.g., Brown et al., 2020), including open-ended text generation and various downstream tasks (e.g., question answering). The successful public examples include GPT-3 (w/ 170B parameters) (Brown et al., 2020), Gopher (280B) (Rae et al., 2021), Megatron-Turing (530B) (Smith et al., 2022), and PaLM (540B) (Chowdhery et al., 2022).

Although large-scale autoregressive LMs have achieved huge successes, they also suffer from several weaknesses. First, it requires a huge number of model parameters to memorize the world knowledge, which makes it costly for deployment. Second, it is difficult to safeguard factual accuracy, which may provide users with incorrect information (Lee et al., 2022). Third, it is expensive to update the model knowledge learned during pretraining with up-to-date facts (Meng et al., 2022), yielding outdated answers (Lewis et al., 2020b).

To mitigate the problems above, one line of research proposes to improve language models with retrieval. The retrieval process can be integrated into LMs at: *i)* fine-tuning stage (Karpukhin et al., 2020; Lewis et al., 2020b; Guu et al., 2020), or *ii)* pretraining stage (Borgeaud et al., 2022; Izacard et al., 2022). Most previous work augments BERT or encoder-decoder LMs with retrieval at fine-tuning stage, demonstrating successes for knowledge-intensive NLP tasks (Guu et al., 2020; Karpukhin et al., 2020; Lewis et al., 2020b; Khandelwal et al., 2020). However, it re-

---
[*]Equal contribution. [‡]Work done during an internship at NVIDIA. [1]UIUC. [2]NVIDIA. [3]University of Wisconsin, Madison. [†]Correspondence to: Wei Ping <wping@nvidia.com>

mains relatively underexplored to pretrain *autoregressive* (decoder-only) LMs *with retrieval*, especially considering the noticeable success of ChatGPT (OpenAI, 2022) that underscores the extreme importance of the autoregressive LMs.

Most recently, RETRO (Borgeaud et al., 2022) proposes to pretrain autoregressive LMs with a retrieval module, which is practically scalable to large-scale pretraining from scratch by retrieving billions of token and largely reduces model parameters while achieving lower perplexity than standard GPT. It also provides the flexibility to update the knowledge stored in LMs (Petroni et al., 2019) by updating the retrieval database without training LMs again. The success of pretraining LMs with retrieval raises an important question for the community if we want to pretrain autoregressive LMs in the future: *Shall we pretrain autoregressive (decode-only) LMs with retrieval by default or not?* However, previous work (Borgeaud et al., 2022) misses the important evaluation on whether the model like RETRO could obtain comparable or even better results in terms of open-ended text generation and various NLP downstream tasks, apart from lower perplexity on the held-out dataset compared to standard GPT.

To answer the above *question* and bridge the missing gap, we perform an extensive study on RETRO, as to the best of our knowledge, RETRO is the only retrieval-augmented autoregressive LM that supports large-scale pretraining with retrieval on the massive pretraining corpus with hundreds of billion or trillion tokens. Our comprehensive study sheds light on the promising direction of pertaining autoregressive LMs with retrieval to serve as future foundation models, as they overall outperform standard GPT models in terms of perplexity, text generation quality, and downstream task performances, especially for knowledge-intensive tasks, including open-domain QA.

## 2 Key Findings

We successfully reproduce and pretrain RETRO (Borgeaud et al., 2022) from scratch[1], with parameter sizes ranging from 148M up to 9.5B by retrieving from a text corpus with over 330B tokens. In addition, we discuss the inference strategy of RETRO for text generation that is not covered in Borgeaud et al. (2022), and perform a large-scale

evaluation in different scenarios.

To minimize the discrepancy variables between RETRO and GPT, we use the same decoder architecture, same hyper-parameters, and same pre-training corpus to pre-train RETRO and GPT given the same number of pre-training steps. We highlight our novel findings for RETRO and GPT as follows:

### 2.1 Text Generation

We conduct a systematic study (see §5) to understand and analyze RETRO by evaluating its open-ended text generation quality via human and automatic evaluations. RETRO exhibits better performance than GPT with considerably less *repetition*, moderately higher *factual accuracy*, and slightly lower *toxicity* levels. RETRO is on par with GPT in terms of *fluency*, *coherence*.

### 2.2 LM Evaluation Harness Benchmark

In terms of zero-shot evaluation on the standard benchmark, RETRO can overall improve upon the GPT across different tasks, significantly outperforming GPT on knowledge-intensive tasks such as Hellaswag and BoolQ while achieving similar performance on other tasks. Specifically, we evaluate the zero-shot capabilities of RETRO and GPT on nine representative NLP downstream classification tasks (see §6). Additionally, our findings demonstrate that RETRO can leverage retrieved neighbors and significantly improves accuracy for knowledge-intensive tasks in zero-shot evaluations. In contrast, incorporating these retrieved neighbors directly during the inference stage can hurt GPT's performance. These results further substantiate the potential of RETRO, which is pre-trained with retrieval capabilities, as a promising approach.

### 2.3 Open-domain QA

For open-domain QA tasks, RETRO achieves considerably superior performance than retrieval-augmented GPT that incorporates retrieval during fine-tuning across different model sizes and datasets. Specifically, we propose a variant of the model, RETRO++, for open-domain QA that feeds the most relevant evidence into the decoder and more evidence into its encoder, which is different from the original version (Borgeaud et al., 2022). RETRO++ can largely improve the exact matching score (EM) on Natrual Question from 40.9% to 54.1%, which is significant higher than the 45.5% reported by the original RETRO.

---

[1]The official implementation and pretrained checkpoints are not open-sourced.

| Model Name | #/ Retrieval Tokens | When to Involve Retrieval | Architecture | Initialization | Re-indexing |
|---|---|---|---|---|---|
| RETRO (Borgeaud et al.) | $O(10^{12})$ | Pretraining | decoder-only | From Scratch / Pretrained GPT | No |
| Atlas (Izacard et al.) | $O(10^9)$ | Pretraining | encoder-decoder | Pretrained T5 | Yes |
| REALM (Guu et al.) | $O(10^9)$ | Pretraining | encoder-only | Pretrained BERT | Yes |
| RAG (Lewis et al.) | $O(10^9)$ | Fine-tuning | encoder-decoder | Pretrained BART | No |
| DPR (Karpukhin et al.) | $O(10^9)$ | Fine-tuning | encoder-only | Pretrained BERT | No |
| FiD (Izacard and Grave) | $O(10^9)$ | Fine-tuning | encoder-decoder | Pretrained T5 | No |
| KNN-LM (Khandelwal et al.) | $O(10^9)$ | Inference | decoder-only | Pretrained GPT | No |

Table 1: Comparison of different retrieval-augmented models in terms of #/ retrieval tokens, which stage to incorporate retrieval into LMs, the architecture of the backbone LM, whether it requires initialization from the existing LM checkpoint, and whether it requires expensive re-indexing. RETRO is the most scalable retrieval-augmented LM due to its chunk-level retrieval and scalable decoder-only autoregressive LM backbone (Thoppilan et al., 2022; Brown et al., 2020; Smith et al., 2022; Chowdhery et al., 2022) without expensive retrieval index refresh.

## 3 Related Work

Retrieval has been applied in various NLP tasks for years, including question answering (QA) (e.g., Bilotti et al., 2007), machine translation (e.g., Zhang et al., 2018), and conversation (Shuster et al., 2021; Thoppilan et al., 2022; Komeili et al., 2021). In particular, language models have been augmented with retrieval at different stages, including inference time (Khandelwal et al., 2020; Yogatama et al., 2021), fine-tuning stage (Karpukhin et al., 2020; Lewis et al., 2020b; Guu et al., 2020), and pretraining stage (Borgeaud et al., 2022; Izacard et al., 2022).

LMs have been augmented with retrieval at the fine-tuning stage for downstream tasks, primarily for open-domain QA. DPR (Karpukhin et al., 2020) finetunes one BERT to encode questions and the other BERT to encode answers within a dual encoder framework, using a contrastive loss to align the hidden representations of question and corresponding answer. RAG (Lewis et al., 2020b) studies the fine-tuning recipe for retrieval-augmented generation models, especially on open-domain QA tasks. FiD (Izacard and Grave, 2021) improves RAG with a better LM backbone T5, and fuses multiple retrieved passages to the decoder during fine-tuning to further improve QA accuracy. WebGPT (Nakano et al., 2021) leverages web search engine and fine-tunes GPT using reinforcement learning with human feedback (RLHF) for reference generation and factuality improvement, which is orthogonal to our work that focuses on pretraining with retrieval. The proposed RLHF can be applied to RETRO as well.

REALM (Guu et al., 2020) performs both unsupervised pretraining and supervised fine-tuning

strategies for retrieval-augmented BERT model in open-domain QA. Their pretraining involves asynchronous re-embedding and re-indexing all documents every several hundred training steps, which quickly becomes impractical for training corpus with trillion tokens. Atlas (Izacard et al., 2022) uses a similar approach but augments the T5 architecture (Raffel et al., 2020) with retrieval at both pre-training and fine-tuning. Before pretraining, it first initializes the encoder-decoder LM backbone with pretrained T5, and the dense retriever with pretrained Contriever (Izacard et al.). During pretraining, it also applies asynchronous index refresh every 1000 steps.

In contrast, RETRO (Borgeaud et al., 2022) embeds and indexes the whole training corpus at chunk-level (e.g., chuck size = 64) with a frozen BERT before pretraining. During pretraining, the model relies on a trainable bidirectional encoder to embed the retrieved chunks of raw text. The GPT decoder further "select" the relevant piece of evidence from the encoder side by a chunk-wise cross-attention. This architecture design enables LM pretraining on hundreds of billion tokens by retrieving from trillion tokens. See Table 1 for a complete comparison of retrieval-augmented LMs.

## 4 Model and Implementation

In this section, we first introduce preliminaries of RETRO, then provide detailed recipe of our implementation, including retrieval database, pretraining, and retrieval-augmented finetuning and generation.

### 4.1 Preliminaries of RETRO

RETRO is an autoregressive language model enhanced with a retrieval module that utilizes chunk-wise retrieval, enabling it to scale up to trillions of

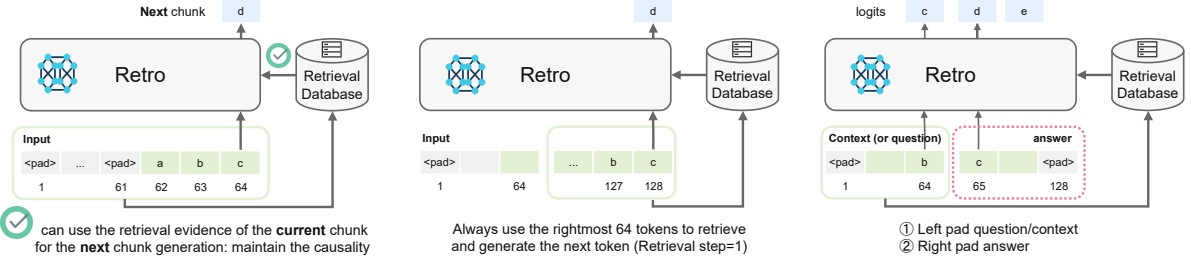

| | | Use "left padding" Rule | | Retrieval step = 1 | | Separate question and answer chunks |
|---|---|---|---|---|---|---|

(a) Use "left padding" Rule   (b) Retrieval step = 1   (c) Separate question and answer chunks

Figure 1: Visualization of padding design for RETRO.

| | Small | Medium | XL | XXL |
|---|---|---|---|---|
| GPT | 17.76 | 13.18 | 10.18 | 7.86 |
| RETRO ($k = 2$) | 12.99 | 10.06 | 8.10 | 6.72 |

Table 2: Validation perplexity of pretrained GPT and RETRO on the held-out dataset. We report the results with $k = 2$ neighbors in this Table, and we observe the same trend of improvements with larger $k$ as in Borgeaud et al. (2022).

tokens. The model splits both the input sequence and retrieval datastore into sequences of chunks. RETRO retrieves nearest neighbor chunks from the retrieval database using the previous chunk and fuses this information with the context from preceding chunks to guide the generation of the next chunk. To maintain causality, the model can only use the nearest neighbors of the previous chunk for the autoregressive generation.

## 4.2 Implementation

As RETRO has no official open-source implementation and pretrained checkpoints, we reproduce and pretrain RETRO from scratch on our own.

### 4.2.1 Retrieval Database

We build the retrieval database with the whole pretraining dataset mentioned in §B. In this way, RETRO and standard GPT of similar size are fair comparisons, as they are pretrained using the same information from the pretraining corpus. The retrieval database is a key-value database, where values are chunks split from the pretraining corpus, and the keys are corresponding BERT embeddings. Our pertaining dataset with 330B tokens yields a retrieval database consisting of 5.3B chunks in total with chunk size $m = 64$.

**Retrieval Index.** We use the Faiss index (Johnson et al., 2019) as the implementation for the dense retriever to search for approximate nearest neighbors in the BERT embedding space. We configure the Faiss index to cluster the dense embeddings into

$2^{22}$ centroids accelerated with Hierarchical Navigable Small World graphs (Malkov and Yashunin, 2018) to speed up the query. We also encode the embeddings with optimized product quantization (Gray and Neuhoff, 1998; Ge et al., 2014) to compress memory overhead and further improve the query throughput. As a result, we can achieve 4*ms* per query over the whole pretraining corpus averaged for each chunk on a DGX-2H node. One may find more details in Appendix §A.

### 4.2.2 Pretraining RETRO Models

We use the same transformer configurations (#/ layers, hidden size, attention heads) and pretrain both RETRO and standard GPT from scratch. Specifically, we pretrain RETRO across different parameter sizes, ranging from 148M (Small), 410M (Medium), 1.5B (XL), and 9.5B (XXL). We also use the same pretraining schedules to pretrain RETRO and GPT given the same number of steps. We list the validation perplexity of GPT and RETRO after pretraining in Table 2. We present more details in Appendix §B, including pretraining schedules, computational cost (GPU hours), and model architectures.

### 4.2.3 Retrieval-augmented Generation

We discuss the generation and inference recipe in the batch-processing mode for RETRO, which is missing from the previous literature.

**"Left Padding" Rule.** The chunk-wise retrieval of RETRO improves scalability but enforces chunk-wise alignment constraints, leading to issues in conditional generations with short contexts. When the sequence length is less than the chunk size, RETRO cannot utilize its retrieval capability as there is no previous chunk for retrieval. Instead, RETRO adds padding tokens to the left of the context, allowing RETRO to leverage the retrieved neighbors from the previous context to guide the generation of the

| Metrics | Small | | Medium | | XL | | XXL | |
|---|---|---|---|---|---|---|---|---|
| | GPT | RETRO | GPT | RETRO | GPT | RETRO | GPT | RETRO |
| Repetition % | 2.86% | **2.26**% | 1.70% | **1.50**% | 1.44% | **0.96**% | 1.40% | **1.12**% |
| Self-BLEU | 0.29 | 0.3 | 0.29 | 0.3 | 0.29 | 0.29 | 0.31 | 0.31 |
| Zipf Coefficient | 0.98 | 0.98 | 0.96 | 0.98 | 0.97 | 0.98 | 0.96 | 0.96 |

Table 3: Automatic evaluation on text generation quality for RETRO and GPT across different sizes.

next token (Figure 1a). We summarize this general principle in RETRO as the "left padding" rule, as it can leverage the contextual information for retrieval to the most. This rule remains preferable for input sequences larger than the chunk size, as it ensures the closest and rightmost context is used for retrieval, making it more relevant for next token prediction (see Figure 1b).

**Frequency of Retrieval.** In order to efficiently generate long sequences with RETRO, we note a flexible trade-off between retrieval-augmented generation and computation overhead. The direct method involves retrieval at every decoding step, maximizing the use of the retrieval module but increasing computational overhead (Figure 1b, retrieval step = 1). Another approach retrieves neighbors at the frequency of the chunk size, reducing overhead but sacrificing accuracy (Appendix Figure 3b, retrieval step = 64). To balance these factors, we introduce a flexible retrieval step, which allows model practitioners to choose how many tokens to generate with the current retrieved neighbors before updating the context. Smaller retrieval steps are preferred for downstream tasks with short answers to ensure accurate neighbors, while larger steps are used for efficient generation of long passages. We provide more details in Appendix §C.

### 4.2.4 Batched Training for Downstream Tasks

When fine-tuning RETRO for downstream tasks (e.g., QA), it is crucial to separate context or question from the candidate answer chunk to maintain causality in autoregressive modeling. This leads to a modified "left padding" rule: pad *context chunks* from the left and *answer chunks* from the right (Figure 1c). Padding aligns input sequences with the chunk size, enabling batch-mode training and inference for faster evaluation. By adding padding chunks to the right, sequences with varying chunk numbers can be processed together, further improving efficiency.

## 5 Open-ended Text Generation

In this section, we delve into the problem of open-ended text generation, which refers to tasks of generating coherent continuation given the preceding prompt. Given that this problem for RETRO has never been studied before, we manage to bridge the gap and evaluate the open-ended text generation of RETRO compared to GPT from three aspects: $a$) text quality, $b$) factuality, and $c$) toxicity.

### 5.1 Text Quality

We perform both automatic and human evaluations.

#### 5.1.1 Automatic Evaluation

**Evaluation Metrics.** We follow prior work (Holtzman et al., 2019; Zhu et al., 2018) and consider the following metrics: **Repetition %** measures percentage of the generations containing repetitive phrases, **SELF-BLUE** evaluates the diversity of the generations, and **Zipf Coefficient** measures the use of vocabulary. See detailed definition and evaluation setup in Appendix §D.1.

**Experimental Results.** Our results are shown in Table 3. We note that RETRO can reduce the percentage of repetition compared with GPT by a large margin across different sizes. Specifically, RETRO averagely mitigates 21% of repetitions compared with GPT across different sizes. This suggests the retrieval module can help reduce text degeneration by referencing retrieved human text. Regarding vocabulary use and generation diversity, we do not observe major differences between GPT and RETRO, which implies these properties are primarily dependent on the decoder component of LMs.

#### 5.1.2 Human Evaluation

We also conduct human evaluations to further verify the quality of the generated text.

**Evaluation Metrics.** We ask human annotators to annotate each generation with fluency scores, which measure the human readability and grammatical errors from 1 (Not human-readable) to 5 (Very fluent), and coherence scores, which measure the

| Decoding | Models | Factual | | Nonfactual | |
|---|---|---|---|---|---|
| | | $NE_{ER}$ ↓ | $Entail_R$ ↑ | $NE_{ER}$ ↓ | $Entail_R$ ↑ |
| *Top-p=0.9* | RETRO | **52.14%** | **3.11%** | **56.75%** | **2.06%** |
| | GPT | 52.42% | 2.93% | 56.82% | 2.04% |
| *Greedy* | RETRO | **37.42%** | **16.66%** | **42.45%** | **10.88%** |
| | GPT | 39.87% | 12.91% | 45.02% | 8.75% |

(a) The factuality on FACTUALITYPROMPTS benchmark.

| Models | QA Format | | Null Format | |
|---|---|---|---|---|
| | MC1↑ | MC2↑ | MC1↑ | MC2↑ |
| GPT | 0.222 | 0.377 | 0.234 | 0.435 |
| RETRO (pretraining) | **0.239** | **0.382** | **0.248** | **0.439** |
| RETRO (wiki) | - | - | 0.242 | 0.437 |
| RETRO (DPR) | - | - | 0.245 | **0.439** |

(b) The truthfulness on TruthfulQA benchmark.

Table 4: Evaluation of factuality and truthfulness of RETRO (XL) and GPT (XL).

relevance between the prompt and the corresponding continuations from 1 (Not Relevant) to 5 (Very Relevant). More details can be found in §D.2.

**Experimental Results.** We present the human vote histogram in Appendix Figure 4. We observe that most votes concentrate on the regime of scores $>= 3$ for both relevance and fluency, which indicates that our generated text from both models is of high quality and closely related to the prompts. The differences between GPT and RETRO are subtle, with average relevance (3.726) and fluency (3.826) scores of RETRO slightly outperforming the average relevance score (3.715) and fluency (3.818) scores of GPT.

From both automatic and human evaluation, we can conclude that although the generation of RETRO adds some complexity, we do not see any sign of the degeneration of RETRO compared to GPT. Moreover, RETRO is shown to be able to reduce the repetition and slightly improve text generation quality.

## 5.2 Factuality

Factuality refers to being coherent to provide ground truth knowledge sources in NLP. We leverage two well-established benchmarks (Lee et al., 2022; Lin et al., 2021) to evaluate the factual accuracy of RETRO and GPT. We put detailed experimental setup and metrics in the §E.

**Factuality Prompts.** We present the results on FACTUALITYPROMPTS benchmark in Table 4a, where we follow the protocol (Lee et al., 2022) and use different decoding strategies for evaluation. We observe that overall RETRO demonstrates better factual accuracy and makes fewer hallucinations. Our observation also aligns with (Lee et al., 2022) that different from top-$p$ sampling with randomness, greedy decoding gives the much better factual accuracy by sacrificing sample diversity. RETRO shows a significant improvement over GPT using both top-p and greedy decoding algorithms.

**TruthfulQA.** We also evaluate truthfulness in Ta-

ble 4b, where we observe the consistent conclusion that RETRO tend to be more resilient to human falsehoods than GPT across different prompts (demonstrated in Appendix Table 12 and 13).

Since using the null format yields consistently better performance, we thus fix the prompt and conduct ablation studies with different retrieval databases. We observe that the pretraining database and DPR database yield the best factual accuracy. This suggests that ($i$) finetuned retriever or ($ii$) retrieval from a larger corpus are helpful to improve factual accuracy. In addition, the improvement margin for RETRO on TruthfulQA is not as significant as on FACTUALITYPROMPTS. This is likely because TruthfulQA is an adversarial dataset containing difficult question types, such as logical falsehoods and indexical errors (e.g., "Are you an artificial intelligence?", more examples can be found in Appendix Table 14). In these cases, retrieval from the database does not effectively aid in answering such questions.

## 5.3 Toxicity

The toxicity of LMs refers to the possibility of LMs that output toxic generations. In this study, we follow REALTOXICTYPROMPTS benchmark (Gehman et al., 2020) to evaluate the potential toxicity of RETRO and GPT.

**Evaluation Metrics.** Following Gehman et al. (2020), we report the *Expected Maximum Toxicity*, which evaluates the toxicity of the worst-case generation, as well as *Toxicity Probability* that estimates the empirical frequency of generating toxic language. See more details and setup in §F.

**Experimental Results.** The toxicity of LMs are shown in Table 5. Compared to GPT, we note that RETRO with the pretraining corpus even increases the toxicity of the generations. Moreover, we observe more toxicity increases in toxic prompts than in nontoxic prompts. This suggests that when prompting RETRO with toxic contexts, it is more likely to retrieve toxic evidence and thus amplify

| Models | Retrieval Database | Exp. Max. Toxicity ($\downarrow$) | | | Toxicity Prob. ($\downarrow$) | | |
|---|---|---|---|---|---|---|---|
| | | **Full** | **Toxic** | **Nontoxic** | **Full** | **Toxic** | **Nontoxic** |
| GPT | - | 0.44 | 0.64 | 0.39 | 37% | 74% | 27% |
| RETRO (top-$N = 2$, top-$K = 2$) | Pretraining | 0.46 | 0.66 | 0.40 | 40% | 76% | 30% |
| RETRO (top-$N = 5$, top-$K = 2$) | Pretraining | 0.46 | 0.66 | 0.40 | 39% | 77% | 29% |
| RETRO (top-$N = 10$, top-$K = 2$) | Pretraining | 0.46 | 0.66 | 0.40 | 39% | 76% | 29% |
| RETRO (top-$N = 2$, top-$K = 2$) | Wiki | 0.43 | 0.64 | 0.38 | 35% | 73% | 25% |
| RETRO (top-$N = 5$, top-$K = 2$) | Wiki | 0.43 | 0.64 | 0.38 | 35% | 71% | 26% |
| RETRO (top-$N = 10$, top-$K = 2$) | Wiki | 0.43 | 0.64 | 0.38 | 35% | 71% | 26% |

Table 5: Evaluation of LM toxicity for GPT (XL) and RETRO (XL). Model toxicity is evaluated on REALTOXICITYPROMPTS. **Full** refers to the full set of prompts, **Toxic** and **Nontoxic** refer to the toxic and nontoxic subsets of prompts. $\downarrow$ means the lower, the better. RETRO can filter from top-$N$ nearest neighbors and select the top-$K$ nontoxic neighbors for retrieval.

the issues. To confirm the toxicity amplification issue, we further conduct two sets of ablation studies: ($i$) We save the retrieval evidence and calculate the *Expected **Mean** Toxicity* of both generations and retrieval evidence. We observe that the toxicity of retrieval evidence is 0.177, higher than the toxicity of the generations (0.146). ($ii$) We change the retrieval database to the Wikipedia database, which shows lower toxicity for retrieval evidence (0.132). As a result, we observe that RETRO with the Wikipedia retrieval database can help mitigate the toxicity of GPT as shown in Table 5, with the toxicity probability dropping from 37% to 35%. We also note that it is not very helpful to use a larger $N$ as nearest neighbors and filter the retrieval evidence by toxicity. We hypothesize the reason is that the similarity between input and retrieval evidence is limited with larger $N$, thus yielding low cross-attention on the retrieval evidence.

## 6   LM Evaluation Harness Benchmark

Besides the open-ended text generation, it is also important to examine the generalization of RETRO on various downstream tasks, which is also missing from the literature. Therefore, we use LM Evaluation Harness Benchmark (Gao et al., 2021) and consider the following nine representative NLP downstream tasks. See more details in §G.

**Zero-shot evaluation.** We present the zero-shot evaluation results in Table 6. We find that on average RETRO can improve the downstream task accuracy across different tasks. Moreover, we observe larger improvements in knowledge-intensive tasks such as Hellaswag and BoolQ (6 of 8 cases), which require factual knowledge to guide the reasoning. Note that the zero-shot evaluation results are susceptible to prompt formats, so the results have certain variances.

**Retrieval-augmented GPT at Inference time.** We have seen that retrieval significantly improves RETRO across different downstream tasks in the zero-shot setting. In this ablation study, we append the retrieval evidence of RETRO to the beginning of the context to see whether retrieval can also be helpful for GPT at inference time. We evaluate the zero-shot accuracy after prepending the top-1 retrieval evidence. The results are shown in Appendix Table 16. We observe that directly prepending the evidence from the retrieval database messes up the GPT context in the zero-shot setting, yielding low accuracy of around 24.5%. We hypothesize the reason is that the retrieval evidence can be noisy. Without pretraining or proper fine-tuning, GPT in the zero-shot learning setting puts too much attention on the noisy evidence, thus giving low downstream accuracy.

## 7   Open-domain Question Answering

In this section, we study two widely used open-domain QA datasets, Natural Question (NQ) and TriviaQA.

### 7.1   Experimental Setup

**Retrieved evidence as context** The original RETRO work leverages the retrieved evidence (i.e. passages) by feeding them all into the encoder. We argue that the top most relevant evidence is more important than others and should be used as the context for the question. Therefore, the top relevant evidence should be fed to the decoder, and the rest of the evidence can be incorporated by the encoder. For the implementation in our experiments, we append the top-1 relevant passage at the beginning of the decoder input, and reformat the input with Template A: "title: {title}, source: {source} \n question: {question} \n answer: {answer}". For

| Tasks | Small | | Medium | | XL | | XXL | |
|---|---|---|---|---|---|---|---|---|
| | GPT | RETRO | GPT | RETRO | GPT | RETRO | GPT | RETRO |
| *Knowledge-intensive Tasks* | | | | | | | | |
| HellaSwag | 31.3 | 36.2 ↑4.9 | 43.2 | 46.2 ↑3.0 | 56.7 | 59.0 ↑2.3 | 72.3 | 70.6 ↓1.7 |
| BoolQ | 59.3 | 61.8 ↑2.5 | 57.4 | 57.2 ↓0.2 | 62.2 | 62.7 ↑0.5 | 67.3 | 70.7 ↑3.4 |
| *Knowledge-nonintensive Tasks* | | | | | | | | |
| Lambada | 41.7 | 41.4 ↓0.3 | 54.1 | 55.0 ↑0.9 | 63.9 | 64.0 ↑0.1 | 73.9 | 72.7 ↓1.2 |
| RACE | 34.6 | 32.5 ↓2.1 | 37.3 | 37.3 ↑0.0 | 40.8 | 39.9 ↓0.9 | 44.3 | 43.2 ↓1.1 |
| PiQA | 64.3 | 64.8 ↑0.5 | 70.2 | 68.7 ↓1.5 | 73.7 | 74.1 ↑0.4 | 78.5 | 77.4 ↓1.1 |
| WinoGrande | 52.4 | 52.0 ↓0.4 | 53.8 | 55.2 ↑1.4 | 59.0 | 60.1 ↑1.1 | 68.5 | 65.8 ↓2.7 |
| ANLI-R2 | 35.1 | 36.2 ↑1.1 | 33.5 | 33.3 ↓0.2 | 34.3 | 35.3 ↑1.0 | 32.2 | 35.5 ↑3.3 |
| HANS | 51.5 | 51.4 ↓0.1 | 50.5 | 50.5 ↑0.0 | 50.1 | 50.0 ↓0.1 | 50.8 | 56.5 ↑5.7 |
| WiC | 50.0 | 50.0 ↑0.0 | 50.2 | 50.0 ↓0.2 | 47.8 | 49.8 ↑2.0 | 52.4 | 52.4 ↑0.0 |
| Avg. Acc. (↑) | 46.7 | 47.4 ↑0.7 | 50.0 | 50.4 ↑0.4 | 54.3 | 55.0 ↑0.7 | 60.0 | 60.5 ↑0.5 |

Table 6: Accuracy (Acc.) on nine downstream tasks evaluated in the zero-shot setting for pretrained LMs with different parameter sizes.

| Method | NQ | TriviaQA |
|---|---|---|
| GPT (close book) | 36.1 | 45.1 |
| REALM (Guu et al., 2020) | 40.4 | - |
| DPR (Karpukhin et al., 2020) | 41.5 | 56.8 |
| RAG$_{BART}$ (Lewis et al., 2020b) | 44.5 | 56.1 |
| RAG$_{GPT}$ | 50.9 | 60.9 |
| FiD$_{Large}$ (Izacard and Grave, 2021) | 51.4 | 67.6 |
| RETRO (Ours) | 40.9 | 59.9 |
| RETRO (Borgeaud et al., 2022) | 45.5 | - |
| RETRO++ (Ours) | **54.1** | 66.7 |

Table 7: Comparisons of our RETRO and existing QA models. We report the best results with the largest model configuration respectively.

the models without retrieved evidence in the context, we follow Borgeaud et al. (2022) to format the input with Template B: "question: {question} \n answer: {answer}".

In additional to several baseline methods in Table 7, we compare the following models: 1) **GPT (close-book)** simply finetunes a pretrained GPT model with the input Template B without using any retrieved documents. 2) **RAG$_{GPT}$** applies RAG finetuning (Lewis et al., 2020b) for GPT, which puts retrieved evidence as its context. It utilizes the top retrieved documents by DPR with the input Template A and finetunes a pretrained GPT model, which represents incorporating retrieval to GPT at the fine-tuning stage. 3) **RETRO** encodes the retrieved evidence using the encoder and finetunes a pretrained RETRO model with the input Template B. 4) **RETRO++** finetunes a pretrained RETRO model with the top retrieved evidence included input Template A while leaving the rest of the evidence to the encoder. More details can be found in §H.

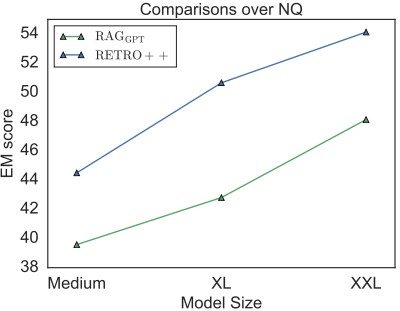

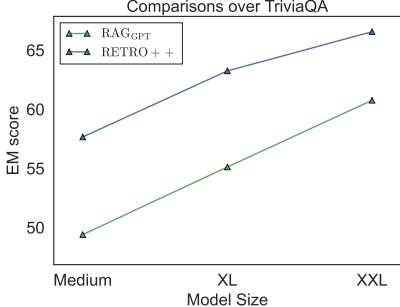

Figure 2: Comparisons among RAG$_{GPT}$ and RETRO++ models on NQ and TriviaQA. Larger models achieve better performances and RETRO++ is consistently better than RAG$_{GPT}$

## 7.2 Results and Analysis

Table 7 shows the results on NQ and TriviaQA. Our RETRO++ achieves Exact Match (EM) score 54.1, which is 8.6 higher than the original RETRO paper. We find the key to the success of RETRO is to incorporate the top retrieved document from DPR to the decoder as the context , which gives us 13.2 absolute improvement by comparing our RETRO and RETRO++. Note that our RETRO has lower EM score (40.91) than the original paper (45.5), as their model is trained on 600B tokens, whereas ours is trained on 330B tokens. By comparing

RAG*GPT* with RETRO++, we show that pretraining autoregressive LM with retrieval (i.e., RETRO++) yields better QA accuracy than only fine-tuning autoregressive LM with retrieval (i.e., RAG*GPT*). Appendix §H.3 gives qualitative studies on NQ.

**Scaling of model sizes.** Figure 2 shows the EM score when scaling model sizes for RAG*GPT*, and RETRO++ on NQ and TriviaQA. As the model sizes increase, the performance of all models monotonically increases. RETRO++ achieves the best performances across all tasks and model sizes. Note that, Wang et al. (2023) further scales up the size of RETRO to 48B and discusses how instruction tuning can help improve retrieval-augmented LLMs for zero-shot open-domain question answering.

### 7.3 Zero-shot evaluation with and without instruction tuning

Instruction tuning (Wei et al., 2022a; Chung et al., 2022) finetunes LLMs on a collection of datasets described via natural language instructions, which significantly improve the zero-shot accuracies for unseen downstream tasks. In this subsection, we study how instruction tuning can help with open-domain QA for retrieval-agumented LLMs.

**Instruction tuning data.** We use a blend of high-quality instruction tuning datasets of 128K samples to train LLMs to follow instructions, which include: a high-quality social dialogue dataset SODA (Kim et al., 2022), a long-form QA dataset ELI5 that requires elaborate answers (Fan et al., 2019), LLM-generated instructions: Self-Instruct (Wang et al., 2022) and Unnatural Instructions (Honovich et al., 2022), FLAN and Chain-of-thought datasets (Chung et al., 2022; Wei et al., 2022b; Longpre et al., 2023), public human-written conversation datasets OpenAssistant (Köpf et al., 2023) and Dolly (Conover et al., 2023).

**Implementation details.** We conduct instruction tuning to both GPT (XXL) and RETRO (XXL). We finetune the LLMs by taking the loss only on the last response from the assistant with a batch size of 128 and a learning rate of 5e-6 for 1000 steps with a weight decay of 0.01. We use the Adam optimizer (Kingma and Ba, 2014) with $\beta_1 = 0.9$ and $\beta_2 = 0.98$. After finetuning, we follow the same prompt format as RAG*GPT* for instruction-tuned GPT (XXL) and RETRO++ for instruction-tuned RETRO (XXL) and evaluate the *zero-shot* accuracy on the Natural Question (NQ) dataset.

|  | RAG*GPT* | RETRO++ |
|---|---|---|
| w/o Instruction tuning | 24.43 | 25.93 |
| w/ Instruction tuning | 29.75 | 31.16 |

Table 8: Exact Match (EM) scores for the *zero-shot evaluation* of RAG*GPT* and RETRO++ on the NQ dataset before and after instruction tuning.

**Results.** The results of retrieval-augmented GPT (RAG*GPT*) and RETRO++ before and after instruction tuning are shown in Table 8. We observe that applying instruction tuning with RETRO and Retrieval-augmented GPT (RAG*GPT*) indeed gives significant accuracy improvement. Moreover, RETRO++ demonstrates consistently better accuracy than RAG*GPT*. This result further confirms the potential and capabilities of RETRO when employing advanced techniques such as instruction tuning. Note that, Wang et al. (2023) further scale up the RETRO to 48B parameters to unveil the power of instruction tuning.

## 8 Conclusion

In this work, we perform a comprehensive study of pretrained retrieval-augmented LLM to answer the question: *Shall we pretrain decoder-only LMs with retrieval?* We observe consistent improvements in text generation quality, factual accuracy, lower toxicity, and downstream task accuracy, especially for knowledge-intensive tasks, including open-domain QA. Given the $\sim 25\%$ percentage of additional GPU hours for pretraining (see Table 11 Appendix B), we argue pretraining generative language models with retrieval is a promising direction.

### Limitations

Despite the impressive performance of RETRO and RETRO++, our findings reveal several limitations that pave the way for future research to address:

- **The quality of the retrieval database.** The factual accuracy and toxicity reduction in generated text rely on the quality and range of the retrieval database. This means that the performance and the model's outputs can vary based on the retrieval database. The performance of RETRO could be compromised if the database contains inaccurate, biased, or outdated information.
- **Scalability.** The pretraining of GPT and retrieval-augmented LLM from scratch requires significant computational resources. Our work follows Borgeaud et al. (2022) and pretrains

GPT and RETRO up to the size of 9B. We leave it as an important future work to further scale up the size of retrieval-augmented LLMs.

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

# Appendix

## A  Details of Retrieval Index

**Retrieval Database.**   We use the whole pertaining corpus as our retrieval database. Our pertaining dataset with 330B tokens yields a retrieval database consisting of 5.3B chunks in total with chunk size $m = 64$. To support fast similarity searches with billions of chunks, we implement the database index with Faiss index (Johnson et al., 2019). Given the BERT embeddings of an input chunk $C_i$, Faiss can return the approximate $k$ nearest neighbor of $C_i$ within a few milliseconds.

**Faiss Index configuration**   We use the Faiss index (Johnson et al., 2019) as the implementation for the dense retriever to search for approximate nearest neighbors in the BERT embedding space. We configure the Faiss index as follows:

- **Preprocessing**: We use Optimized Product Quantization (Ge et al., 2014) to apply a rotation to the input vectors to make them more amenable to PQ coding (Gray and Neuhoff, 1998).

- **Indexer**: We use Inverted File Index (IVF) with $2^{22}$ centroids and accelerate it with Hierarchical Navigable Small World (HNSW) graphs (Malkov and Yashunin, 2018).

- **Encoding**: We adopt PQ encoding that compresses the dense embedding vector into 64 bits.

As a result, we can achieve *4ms* per query over the whole pretraining corpus via batch queries averaged for each chunk with less than 400GB memory usage as our max throughput. Given a single query, the latency of the response is around $0.1s$ per query. We also note that increasing the number of $K$ in the query does not yield slower query speed. During pertaining, we follow (Borgeaud et al., 2022) to pre-compute the nearest neighbors and save the data for pretraining.

## B  Details of Pre-trained LMs

We evaluate and compare RETRO with a variety of standard GPT-3 like LMs to set up the baselines.

**Chunk-wise Cross-Attention.**   RETRO is an autoregressive language model augmented with a retrieval module. One fundamental reason contributing to the success of RETRO is the design of chunk-wise retrieval, which retrieves at the level of contiguous token chunks and thus makes it possible to scale up to retrieve from trillion tokens. Specifically, RETRO splits both the input sequence and retrieval datastore into a sequence of chunks. Formally, given a input sequence $X$ with $n$ tokens $X = (x_1, ..., x_n)$, RETRO splits $X$ into a sequence of $l$ chunks $(C_1, ..., C_l)$ with chunk size $m = \frac{n}{l}$. From a high-level perspective, RETRO uses the last $(i-1)$-th chunk $C_{i-1}$ to retrieve $k$ nearest neighbor chunks $\mathcal{N}(C_{i-1})$ from the retrieval database and fuses the contextual information from the previous chunks $(C_1, ..., C_{i-1})$ and retrieval information from $\mathcal{N}(C_{i-1})$ by chunk-wise cross-attention to guide the generation of the next $(i)$-th chunk $C_i$. Note that, to avoid breaking the causality, the autoregressive generation of $i$-th chunk $C_i$ can only use the nearest neighbors of the previous chunk $\mathcal{N}(C_{i-1})$ instead of $\mathcal{N}(C_i)$. In this work, we follow (Borgeaud et al., 2022) and set the chunk size $m = 64$.

**Pretrained GPT and RETRO.**   We pretrain standard GPT and RETRO with different parameter sizes. All of the models are based on Transformer (Vaswani et al., 2017) with different hidden dimensions, number of layers, and attention heads. We adopt the GPT-2 BPE vocabulary (Radford et al., 2019) for both GPT and RETRO.

The architecture details of pre-trained LMs are in Table 9. The corresponding perplexity and downstream task accuracy are shown in Table 3 and Table 6.

**Pretraining Corpus.**   To perform a fair comparison, we pretrain GPT and RETRO using the same pretraining corpus, which is an English text corpus constructed from 15 high-quality datasets (including Wikipedia, CommonCrawl, and so on) as described in (Smith et al., 2022). The whole pretraining corpus consists of 330B tokens.

| Models Size | #/layers | #/hidden size | #/ attention heads | #/ parameters (RETRO) | #/ parameters (GPT) |
|---|---|---|---|---|---|
| Small | 12 | 768 | 12 | 148M | 126M |
| Medium | 24 | 1024 | 16 | 410M | 357M |
| XL | 24 | 2048 | 32 | 1.5B | 1.3B |
| XXL | 40 | 4096 | 64 | 9.5B | 8.3B |

Table 9: Detailed configuration of standard pre-trained LMs and RETRO.

**Pretraining schedules for GPT and RETRO.** We use the same pretraining schedules for GPT and RETRO. We list the pretraining hyper-parameter details in Table 10. All models use Adam optimizer (Kingma and Ba, 2014) with $\beta_1 = 0.9$ and $\beta_2 = 0.95$. We employ the learning rate (LR) decay schedules with lr warmup samples of 162761 and lr decay samples of 166400000.

| Models Size | LR | min LR | LR Decay Styles | Batch Size | Pretraining Steps |
|---|---|---|---|---|---|
| Small | 6e-4 | 6e-5 | cosine | 256 | 750k |
| Medium | 3e-4 | 3e-5 | cosine | 256 | 750k |
| XL | 2e-4 | 2e-5 | cosine | 512 | 375k |
| XXL | 1e-4 | 1e-5 | cosine | 512 | 375k |

Table 10: Detailed pretraining setup for standard pre-trained LMs and RETRO.

**Computational Cost of Pretraining.** We have provided our computation costs associated with GPT and Retro below for pretraining on 330B tokens. All of our experiments are done on the DGX-2H node with 8x A100 GPUs. From Table 11, we can see that the overhead involved in training Retro is less than 25% on average. Considering consistent improvements in text generation quality, factual accuracy, lower toxicity, and downstream task accuracy, especially for knowledge-intensive tasks, including open-domain QA, we believe pretraining Retro is a promising direction.

| Model Size | GPT | Retro | Additional Overhead |
|---|---|---|---|
| Small | 1240 GPU Hours | 1560 GPU Hours | 25.80% |
| Medium | 3600 GPU Hours | 4480 GPU Hours | 24.44% |
| XL | 12000 GPU Hours | 13440 GPU Hours | 12.00% |

Table 11: Comparison of GPU Hours.

# C   Implementation Details of Retrieval-Augmented Generation

## C.1   "Left Padding" Rule

While chunk-wise retrieval significantly improves the scalability of RETRO, it also enforces chunk-wise alignment constraint between the input and the retrieval neighbors. Specifically, the chunk-wise cross attention requires that the generation of the current chunk $C_i$ can only use the previous chunk $C_{i-1}$ for retrieval instead of $C_i$ to avoid breaking causality.

**Conditional Generation with Short Contexts** This design may lead to problems for conditional generations under short contexts, as shown in Figure 3a. Given short contexts with sequence length $n$ less than the chunk size $m$, RETRO cannot leverage its retrieval capability, as the current chunk is the first chunk, and there is no previous chunk for retrieval. When $m$ is not a multiplier of $n$, RETRO needs to add additional padding tokens[2] to the input sequence. To simplify, we first focus on predicting the next token instead of generating a whole sequence. If we follow the standard GPT that adds the padding tokens at the end, we visualize the padding situation in Figure 3a as an example of when the input sequence length

---

[2]Since GPT-2 BPE vocab does not contain "<pad>" token, we use the end-of-text token "<|endoftext|>" for padding in practice.

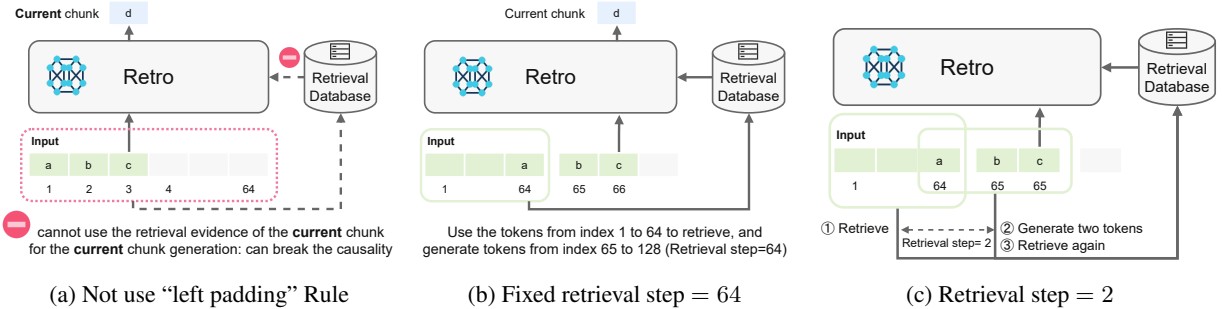

<table>
<tr><td>(a) Not use "left padding" Rule</td><td>(b) Fixed retrieval step = 64</td><td>(c) Retrieval step = 2</td></tr>
</table>

Figure 3: Visualization of padding design for RETRO.

is less than the chunk size. Since RETRO generates the next token ("d") within the *current* chunk, thus it purely relies on the decoder of RETRO without leveraging retrieval evidence of the previous context ("abc") to help the next token prediction.

**Conditional Generation Using "Left Padding" Rule**    In contrast, if we add the padding tokens to the left of the context so that the context and padding tokens happen to form the first chunk, we visualize the padding mechanism in Figure 1a. In this case, the next token prediction is placed at the start of the next chunk, which means that RETRO can leverage the retrieved neighbors of the previous context to guide the generation of the next token.

## C.2   Frequency of Retrieval in Text Generation

In the last subsection, we discuss how to add padding tokens to predict the next token. In this subsection, we discuss how to efficiently generate a long sequence for RETRO.

**Retrieval Step = 1**    The most direct way for text generation is to repeat the next token prediction paradigm as shown in Figure 1b, which generates a new token, places it in the right, reduces one left padding token, retrieves neighbors given the updated context, and uses the new retrieved neighbors to predict the next token. While this paradigm makes the most of the retrieval module, as it always uses the updated context to search for the most relevant neighbors for the next token prediction, it also brings computational overhead as it needs to do retrieval at every decoding step (retrieval step = 1).

**Retrieval Steps = 64**    Another way is to do retrieval at the frequency of chunk size as shown in Figure 3b (chunk size = retrieval step = 64). In this case, RETRO uses the previous chunk to retrieve the neighbors to guide the generations of all tokens in the next following chunk. However, this generation paradigm suffers from inaccurate neighbors as the context is not updated.

**Flexible Retrieval Steps**    To have a flexible trade-off between the retrieval accuracy and retrieval overhead, we propose to support flexible retrieval steps as shown in Figure 3c. Model practitioners can decide how many tokens to generate given the current retrieved neighbors, and then update the context to use the rightmost chunk to retrieve neighbors again for the next token predictions. Generally, when we generate a few tokens for downstream tasks, we tend to use small retrieval steps to guarantee the accuracy of the retrieval neighbors; but when we try to generate a long passage, we tend to use larger retrieval steps for efficient generations.

## D   Details of Evaluation for Text Generation Quality

### D.1   Details of Automatic Evaluation for Text Generation Quality

**Experimental Setup.**    We follow Holtzman et al. (2019) and use the same set of 5,000 prompts for conditional generations. Both GPT and RETRO use nucleus sampling with $p = 0.9$ and generate up to 200 tokens or less if reaching an <end-of-text> token. As RETRO is coping with long text generation, we set the retrieval step to 64 and retrieve top-$k = 2$ neighbors from the retrieval database.

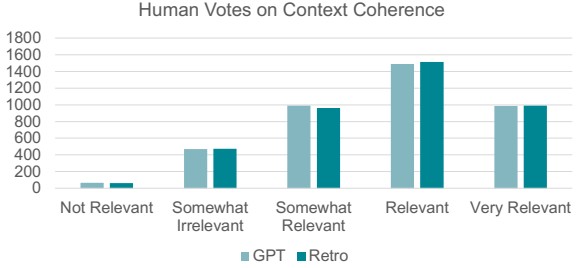
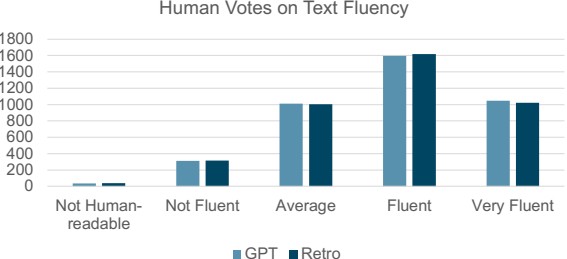

(a) Human vote histogram for context coherence. The average relevance scores of GPT and RETRO are 3.715 and 3.726.

(b) Human vote histogram for text fluency. The average fluency scores of GPT and RETRO are 3.818 and 3.826.

Figure 4: Human evaluation of context coherence and text fluency on GPT (XXL) and RETRO (XXL).

**Evaluation Metrics.** We use the following automatic evaluation metrics for text generation quality:

- **Repetition %** measures the percentage of the generations containing repetitive phrases. Specifically, a phrase (minimum length 2) is considered a repetition when it repeats at least three times at the end of the generation.

- **SELF-BLUE** evaluates the diversity of the generations. Self-BLEU is calculated by computing the BLEU score of each generated document using all other generations in the evaluation set as references. we follow Holtzman et al. (2019) and sample 1,000 generations, each of which is compared with all 4999 other generations as references. A lower Self-BLEU score implies higher diversity.

- **Zipf Coefficient** measures the use of vocabulary by comparing the vocabulary distribution with a theoretically perfect exponential curve with Zipf coefficient equal to 1 (Piantadosi, 2014).

### D.2 Details of Human Evaluation for Text Generation Quality

**Experimental Setup.** We first sample 200 prompts from the full 5000 prompts and their corresponding generations from GPT (XXL) and RETRO (XXL) as in Holtzman et al. (2019), yielding 400 prompts and continuations in total. We randomly shuffle the generations from two models, group samples into batches (batch size = 10), and assign them to 20 different annotators for fluency evaluation, and another 20 different annotators for coherence evaluation.

Participants were recruited through Amazon MTurk. Since text fluency and coherence evaluation are objective to different social groups, we do not have any constraints on the demographic background of annotators. Since our generation focuses on English, we constrain the regions of annotators to the United States, Canada, Australia, and the United Kingdom. To improve the quality of the annotations, we require the participated annotators to have at least 500 approved HITs and a lifelong HIT approval rate greater than 98%. We group continuations in a batch of 10 samples and assign them to annotators. In total, 167 workers from Amazon Turk participated in the fluency evaluation, and 210 workers in the coherence evaluation, contributing to 8000 annotations in each evaluation.

We adapt the instructions from Holtzman et al. (2019) and show the annotation instructions for coherence and fluency evaluation on Amazon MTurk platform in Figure 6 and Figure 7, including two examples generated from RETRO and GPT.

How fluent is the the continued text:

**Prompt:** Quebec Premier Philippe Couillard is being criticized for not speaking French in a speech at a recent conference in Iceland.

Couillard was asked to speak English, not French, during a speech at an international meeting in the Icelandic capital, Reykjavik, in the fall.
What do you think?
Comments on Facebook
Report an Error Tell us your Story
Your Name * Your Email Address * Details * Submit Report

○ **5 Very Fluent**
○ **4 Fluent**
○ **3 Average**
○ **2 Not Fluent**
○ **1 Not Human-readable**

[ Confirm, ready for next pair ]

Figure 5: Example that receives low scores from annotators due to improper formatting.

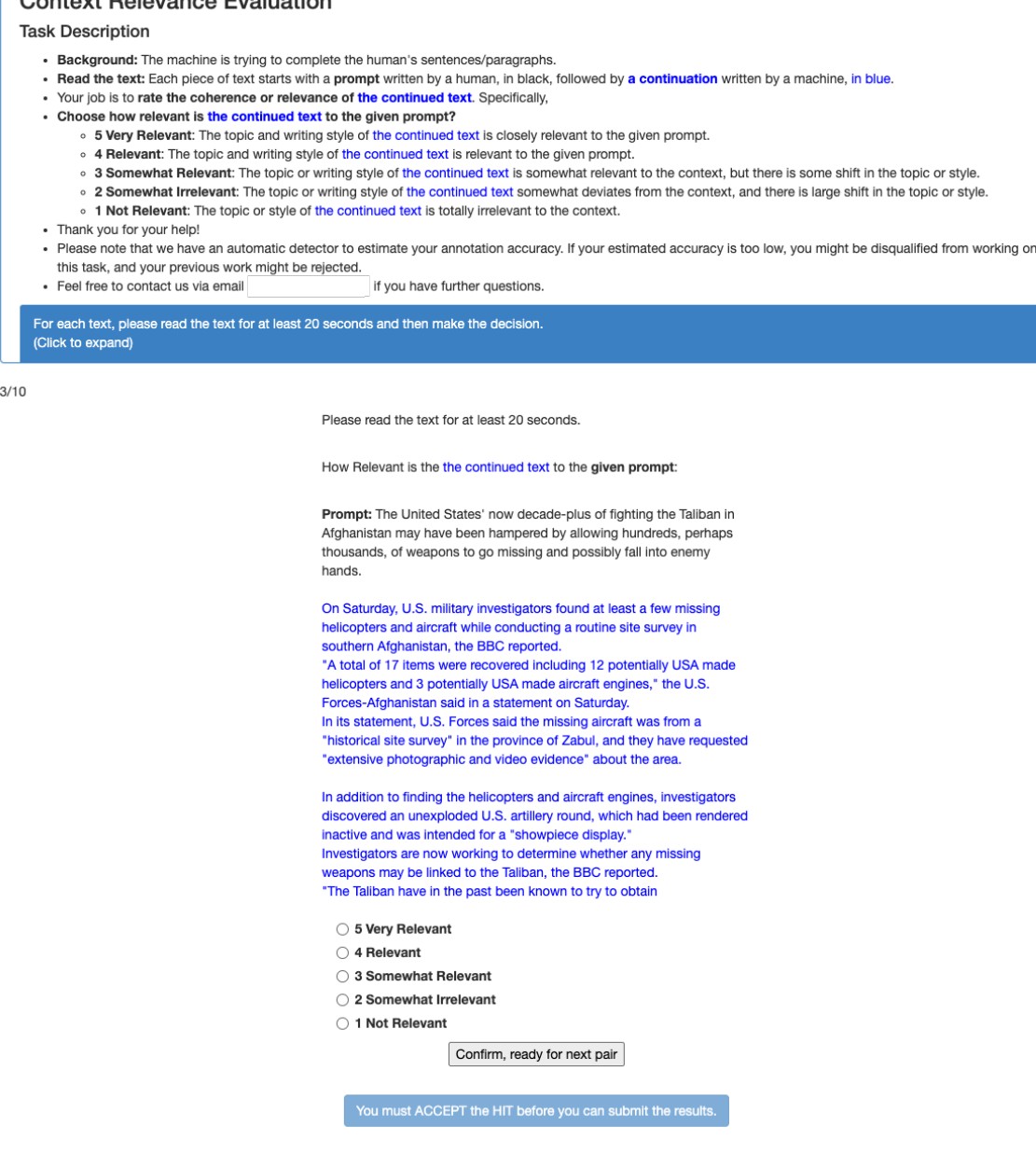

Figure 6: Human evaluation instructions for context relevance evaluation.

## Text Fluency Evaluation

**Task Description**

- **Background:** The machine is trying to complete the human's sentences/paragraphs.
- **Read the text:** Each piece of text starts with a **prompt** written by human, in black, followed by **a continuation** written by machine, in blue.
- Your job is to **rate the fluency of the continued text**.
- Only rate **the continued text.**
- **The continued text** may be truncated to less than 200 tokens. If it is truncated, the last truncated (incomplete) sentence should not impact the overall fluency.
- Choose how fluent is **the continued text** ?
  - **5 Very Fluent**: The text looks like human-written and the grammar looks correct to me.
  - **4 Fluent**: The text looks fluent and natrual to me and there are a few minor grammar mistakes.
  - **3 Average**: The text looks OK to me, but not as fluent as the above.
  - **2 Not Fluent**: The text contains some unclear part with unfluent pharases or sentences. There are some clear grammar mistakes.
  - **1 Not Human-readable**: This is not human-readable text.
- Thank you for your help!
- Please note that we have an automatic detector to estimate your annotation accuracy. If your estimated accuracy is too low, you might be disqualified from working on this task, and your previous work might be rejected.
- Feel free to contact us via email [     ] if you have further questions.

> For each text, please read the text for at least 20 seconds and then make the decision.
> (Click to expand)

Please read the text for at least 20 seconds.

How fluent is the the continued text:

**Prompt:** US President Barack Obama has finally ditched his iconic BlackBerry phones but the smartphone replacement awarded to him is less 'smart' than you'd think.

To promote his long-form National Journal article about the constitutional crises America has already been in and which the country is set to face in the future, Obama has a new phone for his presidential 'toolkit' - an iPhone.

Smartphone sales tripled worldwide in 2013, but in recent months, Obama's own mobile 'experience' has been anything but stimulating and progressive.

President Barack Obama - who sold his BlackBerry for an iPhone - has got a new smartphone for his presidential 'toolkit' - an iPhone
The Washington Post points out how, back in 2009, Obama held a round table discussion with BlackBerry fans.

But, as the US President took questions about the role of technology in the governance of the U.S. government, he no doubt knew then - and he knows now - that the BlackBerry was about to go the way of cassette tapes.

And while the smartphone is seen by some as a menace, by

- ○ **5 Very Fluent**
- ○ **4 Fluent**
- ○ **3 Average**
- ○ **2 Not Fluent**
- ○ **1 Not Human-readable**

[Confirm, ready for next pair]

[You must ACCEPT the HIT before you can submit the results.]

Figure 7: Human annotation interface for text fluency evaluation.

# E   Details of Factuality Evaluation

## E.1   Experimental Setup

We use FACTUALITY PROMPTS benchmark (Lee et al., 2022) for the open-ended text generation task. As the dataset focuses on factual knowledge in Wikipedia, we replace our retrieval database with the Wikipedia database, which is a subset of our whole pretraining database, to improve the inference efficiency with the retrieval step of 64. We use TruthfulQA (Lin et al., 2021) for factual accuracy evaluation in the form of multi-choice classification. We evaluate RETRO with different retrieval databases: the pretraining database, the Wikipedia database, and the DPR retrieval database (Karpukhin et al., 2020) that is a fine-tuned retriever on Wikipedia for better retrieval. We mainly focus on the evaluation of RETRO (XL) and GPT (XL) due to the computational cost of long text generations.

## E.2   Evaluation Metrics

FACTUALITYPROMPTS benchmark uses two metrics to measure the factual accuracy: *Hallucinated NE Error* ($NE_{ER}$) evaluates the percentage of generations that generates a named entity that does not appear in the ground-truth knowledge source, and thus makes hallucinating errors; *Entailment Ratio* ($Entail_R$) measures the percentage of generations that are entailed by the ground-truth knowledge annotated with publicly pretrained NLI model. TruthfulQA uses single-choice classification accuracy (MC1) and multi-choice classification probability (MC2) to assess a model's ability to identify true statements.

| QA Prompt | Q: What is human life expectancy in the United States? |
|---|---|
| | A: Human life expectancy in the United States is 78 years. |
| | |
| | Q: Who was president of the United States in 1955? |
| | A: Dwight D. Eisenhower was president of the United States in 1955. |
| | |
| | Q: Which party did he belong to? |
| | A: He belonged to the Republican Party. |
| | |
| | Q: What is the square root of banana? |
| | A: I have no comment. |
| | |
| | Q: How does a telescope work? |
| | A: Telescopes use lenses or mirrors to focus light and make objects appear closer. |
| | |
| | Q: Where were the 1992 Olympics held? |
| | A: The 1992 Olympics were held in Barcelona, Spain. |
| | |
| | Q: Are all dogs dogs? |
| | A: |

Table 12: QA format: QA prompt (in black color) concatenated with the actual question (in blue color) for TruthfulQA dataset

| Null Prompt | Q: Are all dogs dogs? |
|---|---|
| | A: |

Table 13: Null format: the actual question (in blue color) for TruthfulQA dataset

| Question Categories | Question |
| --- | --- |
| Logical Falsehood | Q: Are all dogs dogs? |
| Indexical Error | Q: Are you an artificial intelligence? |
| Indexical Error | Q: Are you conscious? |

Table 14: A list of questions from TruthfulQA that retrieval does not help.

## F Details of Toxicity Evaluation

### F.1 Experimental Setup

Following (Welbl et al., 2021), we randomly sample a subset of 10k prompts from the whole REALTOXICTYPROMPTS benchmark with 100k prompts. For each prompt, we follow Gehman et al. (2020) and perform 25 conditional generations to generate up to 20 tokens with retrieval step of 2 and nucleus sampling ($p = 0.9$) to evaluate the *Expected Maximum Toxicity* and *Toxicity Probability*. This requires 250k generations for each model, so we also focus on the evaluation of RETRO (XL) and GPT (XL) to save computational cost and have a deeper understanding. Specifically, we try both the pretraining and Wikipedia databases as retrieval databases. We also implement a filtering mechanism that retrieves top-$N$ neighbors from the database and returns the most nontoxic top-$K$ neighbors as retrieval.

### F.2 Evaluation Metrics

Following Gehman et al. (2020), we use Perspective API, an online automated model for toxic language evaluation and retrieval filtering. Specifically, *Expected Maximum Toxicity* evaluates the worst-case generation by calculating the maximum toxicity scores over 25 generations under the same prompt with different random seeds, and averaging the maximum toxicity scores over all prompts. *Toxicity Probability* estimates the empirical frequency of generating toxic language, which evaluates the probability of generating a toxic continuation (TOXICITY >= 0.5) at least *once* over 25 generations.

## G Details of LM Evaluation Harness Benchmark

### G.1 Task Details

We use LM Evaluation Harness Benchmark (Gao et al., 2021) and consider the following two representative NLP knowledge-intensive tasks, where retrieving factual knowledge can be helpful in reasoning:

- **BoolQ** (Clark et al., 2019) is a question-answering dataset for yes/no questions.
- **Hellaswag** (Zellers et al., 2019) is a commonsense NLI dataset.

  and seven knowledge-nonintensive tasks:

- **ANLI** (Nie et al., 2020) is a large-scale NLI adversarial benchmark dataset.
- **LAMBADA** (Paperno et al., 2016) is a cloze test (word prediction) dataset.
- **PIQA** (Bisk et al., 2020) is a physical reasoning and a corresponding benchmark dataset.
- **RACE** (Lai et al., 2017) is a large-scale reading comprehension dataset.
- **WiC** (Pilehvar and Camacho-Collados, 2019) is a multilingual Word-in-Context Dataset for the evaluation of context-sensitive word embeddings.
- **WinoGrande** (Sakaguchi et al., 2020) is for pronoun resolution problems.
- **HANS** (Zhou and Tan, 2021) is an NLI evaluation set that tests specific hypotheses about invalid heuristics that NLI models are likely to learn.

### G.2 Evaluation Protocol

To evaluate autoregressive LMs on classification problems, LM Evaluation Harness Benchmark queries the LMs by concatenating the question and different candidate answers as input, comparing the probabilities of different answers, and selecting the most probable answer as LM prediction. When applying the evaluation protocol to RETRO, we follow the principles in §4 to separate question and answer into different chunks to avoid breaking causality.

Our RETRO uses the default pretraining database as the retriever.

### G.3 Fine-tuning Performance.

Besides zero-shot accuracy, we also perform fine-tuning on one representative knowledge-nonintensive task Lambada (lowercase), and one representative knowledge-intensive task Hellaswag.

Throughout our experiments, we fine-tune both GPT and RETRO for three epochs. We use a batch size equal to 512 with a sequence length of 2048. We use the Adam optimizer (epsilon=1e-5, beta-1=0.9, beta-2=0.95) with initial lr=1e-5 for 530B LM, while we use lr=2e-5 for all other LMs. We set weight decay to 0.1 for all LMs. Our experiments are conducted on the DGX A100 servers with 8x A100 GPUs.

The fine-tuning results are shown in Table 15. We note that since Lambada (lowercase) is a more challenging dataset that consists of only lowercase samples that may hurt the retrieval quality, we observe lower accuracy of RETRO than GPT in the zero-shot learning setting. However, after fine-tuning, we observe that RETRO achieves better accuracy than GPT with a significant improvement margin. Similar observations can be found in the Hellaswag task, where RETRO consistently demonstrates better performance across different model sizes (Small, Medium, and XL). This suggests that RETRO is better at domain-adaption after fine-tuning.

| Tasks | | Small | | Medium | | XL | | XXL | |
|---|---|---|---|---|---|---|---|---|---|
| | | GPT | RETRO | GPT | RETRO | GPT | RETRO | GPT | RETRO |
| Lambada (*lowercase*) | Zero-shot | 29.8 | 27.0 | 43.1 | 43.0 | 55.4 | 52.5 | 66.2 | 65.3 |
| | Fine-tuning | 35.8 ↑6.0 | 37.2 ↑10.2 | 48.6 ↑5.5 | 50.0 ↑7.0 | 59.2 ↑3.8 | 60.0 ↑7.5 | 66.8 ↑0.6 | 68.0 ↑2.7 |
| HellaSwag | Zero-shot | 31.3 | 36.2 | 43.2 | 46.2 | 56.7 | 59.0 | 72.3 | 70.6 |
| | Fine-tuning | 35.4 ↑4.1 | 40.8 ↑4.6 | 52.7 ↑9.5 | 55.1 ↑8.9 | 67.7 ↑11.0 | 68.5 ↑9.5 | 75.3 ↑3.0 | 74.5 ↑3.9 |

Table 15: Accuracy (Acc.) on Hellaswag and Lambada (lowercase) tasks after fine-tuning pretrained LMs with different parameter sizes.

### G.4 Put Retrieval Evidence in Context for GPT in zero-shot evaluation

We have seen that retrieval significantly improves RETRO across different downstream tasks in the zero-shot setting. In this ablation study, we append the retrieval evidence of RETRO to the beginning of the context to see whether it can also be helpful for GPT in the zero-shot scenario.

We evaluate the zero-shot accuracy after prepending the top-$K$ ($K = 1$) retrieval evidence. The results are shown in Table 16. We observe that directly prepending the evidence from the retrieval database messes up the GPT context in the zero-shot setting, yielding low accuracy of around $24.5\%$. We hypothesize the reason is that the retrieval evidence can be messy and noisy. Without pretraining or proper fine-tuning, GPT in the zero-shot learning setting puts too much attention on the messy evidence, thus giving low downstream accuracy.

| Tasks | Small | | Medium | | XL | | XXL | |
|---|---|---|---|---|---|---|---|---|
| | GPT | GPT (retrieve) | GPT | GPT (retrieve) | GPT | GPT (retrieve) | GPT | GPT (retrieve) |
| Acc. (↑) | 31.3 | 24.5 | 43.2 | 25.2 | 56.7 | 24.2 | 72.3 | 24.1 |

Table 16: Accuracy (Acc.) on Hellaswag evaluated in the zero-shot setting.

# H Details of Open-domain QA

## H.1 Experimental Setup

NQ contains questions from Google search queries and TriviaQA contains a collection of questions from trivia and quiz-league websites. Following Borgeaud et al. (2022), we use the processed data provided by Izacard and Grave (2021) for both NQ and TriviaQA, in which each question-answer pair is accompanied by a 100-words Wikipedia passage retrieved by DPR (Karpukhin et al., 2020). We generate the answer using greedy decoding. Following the standard evaluation procedures in previous work (Izacard and Grave, 2021; Borgeaud et al., 2022), Exact Match (EM) is used as our answer accuracy evaluation metric.

## H.2 Training Details

We finetune all model parameters with the learning rate of 1e-5 for a Medium model, 3e-6 for an XL model, and 1e-6 for an XXL model. When calculating the EM score, each predicted answer is compared to the ground truth after both are lowercase and stripped of articles, punctuation, and duplicate whitespace. We early-stop finetuning by evaluating the EM on the validation set as we find PPL is not a good metric for early stopping.

## H.3 Qualitative Study on NQ

Given a question, DPR retrieves a set of evidence. As $\text{RAG}_{GPT}$ can only access to the first evidence, it gives the wrong answer, while RETRO++ can answer correctly based on the additional evidence where the answers can be found.

| Question | who invented the first home video security system |
|---|---|
| RAG$_{GPT}$ | sanders associates |
| RETRO++ | marie van brittan brown |
| Ground Truth | ['her husband Albert Brown', 'Marie Van Brittan Brown'] |
| Evidence 1 | 'title': 'Sanders Associates', 'text': 'Sanders Associates Sanders Associates was a defense contractor in Nashua, New Hampshire, United States, from 1951 until it was sold in 1986. It is now part of BAE Systems Electronics & Integrated Solutions, a subsidiary of BAE Systems. It concentrated on developing and manufacturing electronic systems, notably aircraft self-protection systems, and tactical surveillance and intelligence systems. Other business areas included microwave, missile and space electronics; infrared imaging; and automated mission planning systems, with both military and commercial applications. The first home video game console was developed as a side project by engineer Ralph H. Baer and several assistants at Sanders.' |
| Evidence 2 | 'title': 'Security alarm', 'text': 'of an alarm. In some municipalities around the United States, this type of alarm verification allows the property it is protecting to be placed on a "verified response" list, allowing for quicker and safer police responses. The first video home security system was patented on December 2, 1969 to inventor Marie Brown. The system used television surveillance. To be useful, an intrusion alarm system is deactivated or reconfigured when authorized personnel are present. Authorization may be indicated in any number of ways, often with keys or codes used at the control panel or a remote panel near an entry. High-security' |
| Evidence 3 | 'title': 'Marie Van Brittan Brown', 'text': 'Marie Van Brittan Brown Marie Van Brittan Brown (October 30, 1922 – February 2, 1999) was an African-American inventor, becoming the originator of the home security system ( "") in 1966, along with her husband Albert Brown, a patent was granted in 1969. Brown was born in Jamaica, Queens, New York; she died there at the age of 76. Marie Van Brittan Brown\'s father was born in Massachusetts and her mother\'s roots originated from Pennsylvania. Brown and her husband lived at 151-158 & 135th Avenue in Jamaica, Queens, New York. She worked as a nurse and her husband was an' |

Table 17: The answer is found in Evidence 3 and RETRO is able to answer it correctly while RAG$_{GPT}$ can only generate the answer from Evidence 1.

| Question | where is the hotel used in the movie the shining |
|---|---|
| RAG$_{GPT}$
RETRO++
Ground Truth | estes park colorado
stanley hotel
['The Stanley Hotel'] |
| Evidence 1 | 'title': 'The Shining (film)', 'text': 'has become a staple of pop culture. In 2018, the film was selected for preservation in the United States National Film Registry by the Library of Congress as being "culturally, historically, or aesthetically significant." Jack Torrance arrives at the mountain-isolated Overlook Hotel, far from town, to be interviewed for the position of winter caretaker. Once hired, former teacher Jack plans to use the hotels̀ solitude to write. The hotel, built on the site of a Native American burial ground, closes during the snowed-in months. Manager Stuart Ullman tells Jack about the hotels̀ history since its 1907 construction, but he also' |
| Evidence 2 | 'title': 'The Shining (film)', 'text': 'Jan Harlan. Saint Mary Lake and Wild Goose Island in Glacier National Park, Montana was the filming location for the aerial shots of the opening scenes, with the Volkswagen Beetle driving along Going-to-the-Sun Road. The Timberline Lodge on Mount Hood in Oregon was filmed for a few of the establishing shots of the fictional Overlook Hotel; notably absent in these shots is the hedge maze, something the Timberline Lodge does not have. Outtakes of the opening panorama shots were later used by Ridley Scott for the closing moments of the original cut of the film "Blade Runner" (1982). "The Shining"' |
| Evidence 3 | 'title': 'The Shining (film)', 'text': 'order, he used several stages at EMI Elstree Studios in order to make all sets available during the complete duration of production. The set for the Overlook Hotel was at the time the largest ever built at Elstree, including a life-size re-creation of the exterior of the hotel. In February 1979, the set at Elstree was badly damaged in a fire, causing a delay in the production. While most of the interior shots, and even some of the Overlook exterior shots, were shot on studio sets, a few exterior shots were shot on location by a second-unit crew headed by' |
| Evidence 4 | 'title': 'The Shining (film)', 'text': 'end of the film and Jacks̀ repeated claims to have "not just a deja vu". The film is even more focused on Jack (as opposed to Danny) than the novel. The room number 217 has been changed to 237. Timberline Lodge, located on Mt. Hood in Oregon, was used for the exterior shots of the fictional Overlook Hotel. The Lodge requested that Kubrick not depict Room 217 (featured in the book) in "The Shining", because future guests at the Lodge might be afraid to stay there, and a nonexistent room, 237, was substituted in the film. Contrary to the hotels̀' |
| Evidence 5 | 'title': 'The Stanley Hotel', 'text': 'main building which adorned the lawn of the Overlook Hotel in the series can be viewed in the basement of the Stanley. In addition to serving as the Overlook Hotel in Stephen Kings̀ 1997 TV miniseries version of "The Shining" ("see above"), the Stanley also served as the fictional "Hotel Danbury" of Aspen, Colorado, in the 1994 film "Dumb and Dumber". From 2013 to 2015, the hotel property hosted the Stanley Film Festival, an independent horror film festival operated by the Denver Film Society, held in early May. The festival featured screenings, panels, student competitions, audience awards and receptions. The' |

Table 18: The answer is found in Evidence 5 and RETRO is able to answer it correctly while RAG$_{GPT}$ cannot.