# OpenReview forum: "Shall We Pretrain Autoregressive Language Models with Retrieval? A Comprehensive Study"
_EMNLP/2023/Conference — EMNLP 2023 Main_

### Official Review · Reviewer_HmWG · 2023-08-07

**Soundness:** 4

**Excitement:**

3: Ambivalent: It has merits (e.g., it reports state-of-the-art results, the idea is nice), but there are key weaknesses (e.g., it describes incremental work), and it can significantly benefit from another round of revision. However, I won't object to accepting it if my co-reviewers champion it.

**Missing References:**

missing reference: https://arxiv.org/abs/2305.11564

**Paper Topic And Main Contributions:**

In this paper, the authors delve into the question of whether we should pretrain large autoregressive LMs with retrieval. To address this question, the authors reproduce a scalable pretrained retrieval-augmented LM, Retro, and conduct a comprehensive study revealing the following novel findings:
- RETRO outperforms GPT on text generation with much less degeneration (i.e., repetition), moderately higher factual accuracy, and slightly lower toxicity with a nontoxic retrieval database.
- On the LM Evaluation Harness benchmark, RETRO largely outperforms GPT on knowledge-intensive tasks, but is on par with GPT on other tasks.

Furthermore, the authors propose a variant of the model, RETRO++, which significantly enhances the performance of the open domain QA task.

**Questions For The Authors:**

Q1: According to the experimental results, we can see that for larger models, pre-training with retrieval leads to smaller improvement, so is this really necessary for LLMs with more than 100b parameters?

Q2: I believe that the effectiveness of "Retrieval-augmented GPT at Inference time" might be limited due to its lack of instruction fine-tuning. Based on my experience using ChatGPT on a daily basis, I've noticed that retrieving relevant text initially has led to improvements in the results. Could you please elaborate further on this concept or perhaps provide additional experimental results to support it?

Q3: Will inconsistent retrieval databases used for inference and training have an impact on model performance? If so, would this hinder the ability to update the LM's knowledge?

Q4: The results in Table 6 seem to be unstable and I wonder how robust the model is.

**Reasons To Accept:**

The authors reproduce RETRO and conduct extensive experiments comparing with GPT, yielding novels findings. Their work highlights the promising potential of pre-training language models with retrieval.

**Reasons To Reject:**

- The method of this paper may lack some novelty, as it follows RETRO which is proposed by others. Furthermore, the conclusions drawn from the experiment seem to lack originality.
- The experimental results are not sufficient to demonstrate that pretraining with retrieval is a more efficient method. (details in Questions)
- I think the author's comparison of solely RETRO to a base model like GPT to be insufficient. Recently, LLMs are often subject to instruction tuning or RLHF training. I am particularly intrigued by how RETRO's performance stacks up against such models, and I wonder the extent of RETRO's capabilities when employing such advanced techniques.

**Reproducibility:**

3: Could reproduce the results with some difficulty. The settings of parameters are underspecified or subjectively determined; the training/evaluation data are not widely available.

**Reviewer Confidence:**

4: Quite sure. I tried to check the important points carefully. It's unlikely, though conceivable, that I missed something that should affect my ratings.

---

> ### Author Rebuttal · Authors · 2023-08-29
>
> We thank the reviewer for the valuable comments and feedback. We are glad that the reviewer finds our work demonstrating **novel findings** through extensive experiments and highlighting the promising potential of pretraining LLMs with retrieval. We provide detailed responses below.
>
> > 1. “The method of this paper may lack some novelty, as it follows RETRO which is proposed by others. Furthermore, the conclusions drawn from the experiment seem to lack originality.”
>
>   - We would like to clarify that all the conclusions and findings drawn from our experiments are original and have not been discussed in previous work. Although we follow the original RETRO paper (so did many follow-up works of the GPT family), our work provides **novel findings** that shed light on the intrinsic properties of Retro in text generation, downstream tasks, and fine-tuning capabilities. We believe novelty can refer to not only novel models, but also novel findings.  Our extensive experiments demonstrate that there are consistent improvements in text generation quality (less degeneration, higher factual accuracy, and lower toxicity), and downstream task accuracy (especially for knowledge-intensive tasks, including open-domain QA with the simple yet effective RETRO++ architecture) without significant overhead in pretraining.
>
>   - Moreover, given the importance of recent autoregressive LLMs that are expensive to pretrain, our work manages to answer an important and practical problem on whether we should pre-train decoder-only LLMs with retrieval. Our work conveys a clear signal to scale up the model sizes and pretrain larger autoregressive LMs with retrieval as a promising future direction.
>
>
> > 2. “The experimental results are not sufficient to demonstrate that pretraining with retrieval is a more efficient method. (details in Questions): I believe that the effectiveness of "Retrieval-augmented GPT at Inference time" might be limited due to its lack of instruction fine-tuning. Based on my experience using ChatGPT on a daily basis, I've noticed that retrieving relevant text initially has led to improvements in the results. Could you please elaborate further on this concept or perhaps provide additional experimental results to support it? / I think the author's comparison of solely RETRO to a base model like GPT to be insufficient. Recently, LLMs are often subject to instruction tuning or RLHF training. I am particularly intrigued by how RETRO's performance stacks up against such models, and I wonder the extent of RETRO's capabilities when employing such advanced techniques.”
>
>   - Thank you for constructive feedback. We follow your suggestion and add instruction tuning to both GPT (XXL) and Retro (XXL) to demonstrate the effectiveness of Retro and retrieval-augmented GPT. Specifically, we construct a blend of instruction tuning datasets consisting of 128K training samples from the [soda dataset](https://huggingface.co/datasets/allenai/soda)[1], [ELI5 dataset](https://github.com/facebookresearch/ELI5) [2], [FLAN dataset](https://github.com/google-research/FLAN) [3], [Open Assistatant dataset](https://huggingface.co/datasets/OpenAssistant/oasst1) [4], etc. We fine-tune both GPT and Retro on the instruction tuning dataset and evaluate the zero-shot accuracy on the Natural Question (NQ) dataset (Note: the training data of NQ is not in our instruction tuning dataset). The results of retrieval-augmented GPT ($\text{RAG}_\text{GPT}$) and Retro++ before and after instruction tuning are shown below:
>
> | Zero-shot evaluation on NQ  | $\text{RAG}_\text{GPT}$ (XXL)    | Retro++ (XXL)  |
> | ------------------------------ | ------ | ------ |
> | w/o Instruction tuning     | 0.2443 | **0.2593** |
> | w/ Instruction tuning | 0.2975 | **0.3116** |
>
>   - From the table above, we observe that stacking up instruction tuning with Retro and Retrieval-augmented GPT indeed gives significant accuracy improvement. Moreover, Retro demonstrates consistently better accuracy than retrieval-augmented GPT. This result further confirms the potential and capabilities of Retro when employing advanced techniques like instruction tuning.
>
> [1] Kim, Hyunwoo, Jack Hessel, Liwei Jiang, Ximing Lu, Youngjae Yu, Pei Zhou, Ronan Le Bras, Malihe Alikhani, Gunhee Kim, Maarten Sap and Yejin Choi. “SODA: Million-scale Dialogue Distillation with Social Commonsense Contextualization.” ArXiv abs/2212.10465 (2022): n. Pag.
>
> [2] Fan, Angela, Yacine Jernite, Ethan Perez, David Grangier, Jason Weston and Michael Auli. “ELI5: Long Form Question Answering.” ArXiv abs/1907.09190 (2019): n. Pag.
>
> [3] Wei, Jason, Maarten Bosma, Vincent Zhao, Kelvin Guu, Adams Wei Yu, Brian Lester, Nan Du, Andrew M. Dai and Quoc V. Le. “Finetuned Language Models Are Zero-Shot Learners.” ArXiv abs/2109.01652 (2021): n. Pag.
>
> [4] Kopf, Andreas et al. “OpenAssistant Conversations - Democratizing Large Language Model Alignment.” ArXiv abs/2304.07327 (2023): n. pag.
>
>
> > 3. “According to the experimental results, we can see that for larger models, pre-training with retrieval leads to smaller improvement, so is this really necessary for LLMs with more than 100b parameters?”
>
>   - We would like to note that from Figure 2, the improvement of Retro on knowledge-intensive tasks, including both NQ and TriviaQA datasets, is still significant across models of different sizes. In Figure 2, we do not see any sign of diminishing margin for the improvement. Thus, we believe that it is promising to scale up the model sizes and pretrain large LMs as an important future direction. We treat it as an important future work.
>
>
> > 4. “Will inconsistent retrieval databases used for inference and training have an impact on model performance? If so, would this hinder the ability to update the LM's knowledge?”
>
>   - Inconsistent retrieval databases used in inference and training may not hurt the model performance but improve it instead, which is one of the advantages and flexibility that Retro can bring to update LLM’s knowledge. For example, during inference, we replace the default pretraining database with the Wiki database in our toxicity experiments, which reduces the toxicity of retrieved evidence and thus yields lower toxicity in LLM generations. During training, we replace the default pretraining database with Wikipedia passages retrieved by DPR used in standard QA benchmarks, which yields higher accuracy among the state-of-the-art QA models.
>
>
> > 5. “The results in Table 6 seem to be unstable and I wonder how robust the model is.”
>
>   - We agree that zero-shot evaluation on the LM Evaluation Harness benchmark can be unstable and sensitive to the prompts for the specific tasks (which is known among LLM researchers). Therefore, we conduct the zero-shot evaluation across **nine different tasks**, including 2 knowledge-intensive tasks and 7 knowledge-nonintensive tasks, to demonstrate the accuracy of Retro and GPT. While the average accuracy of Retro across different knowledge-nonintensive tasks is close to GPT, we observe that the average accuracy of Retro across knowledge-intensive tasks is indeed significantly better than GPT, surpassing GPT by 1.84% on average for models of different sizes.
>
> > 6. Missing reference : https://arxiv.org/abs/2305.11564
>   - Many thanks for the reference. This paper introduces PlugLM to pretrain LMs with a differentiable plug-in retrieval module. The authors have applied PlugLM to BERT-based encoder-only models, and demonstrate its capabilities and potential in domain adaptation, knowledge update, and in-task knowledge learning. It would be an interesting future direction to apply the technique to autoregressive decoder-only LMs. We will add the reference and include more discussion in the revision.

---

### Official Review · Reviewer_jg92 · 2023-08-11

**Soundness:** 4

**Excitement:**

4: Strong: This paper deepens the understanding of some phenomenon or lowers the barriers to an existing research direction.

**Paper Topic And Main Contributions:**

This paper investigates whether retrieval helps in pre-training autoregressive LMs. To this end, the authors first reproduce and pretrain RETRO, and then compare the performance of RETRO with GPT on open-ended text generation,  LM Eval Harness Benchmark, and open-domain QA.

The findings are that 1) RETRO outperforms GPT on text generation with less repetition, moderately higher factual accuracy, and
slightly lower toxicity with a nontoxic retrieval database. 2) On the LM Evaluation Harness benchmark, RETRO largely outperforms GPT on knowledge-intensive tasks, but is on par with GPT on other tasks. 3) On the QA tasks, it is helpful to incorporate the top retrieved document from DPR  to the decoder as the context. All these findings more or less align with my intuition, so nothing surprising. The experiments are relatively comprehensive, but I think the authors should add a simple baselines which is GPT+retrieval in the few-shot setting.

**Questions For The Authors:**

1. In Table 5 are the differences statistically significant enough?


**Reasons To Accept:**

Relatively comprehensive investigation on an important topic: retrieval-augmented LM. I appreciate that the authors 1) reproduced an retrieval-based LM whose implementation is not open-sourced, based on which the authors later made small improvements, and 2) conducted experiments on a wide range of tasks.

**Reasons To Reject:**

Too few baseline models are included in text-generation (section 5) and the LM evaluation harness benchmark (section6). Since the vanilla GPT is not aided by retrievals, it is not surprising to see it performs worse on knowledge-intensive tasks. The authors said "We observe that directly prepending the evidence from the retrieval database messes up the GPT context in the zero-shot setting", but a straightforward alternative is to write an instruction with few-shot examples.

I would suggest the authors to include a few retrieval-based baselines, too, such as GPT + retrieval at inference time.

**Reproducibility:**

3: Could reproduce the results with some difficulty. The settings of parameters are underspecified or subjectively determined; the training/evaluation data are not widely available.

**Reviewer Confidence:**

3: Pretty sure, but there's a chance I missed something. Although I have a good feel for this area in general, I did not carefully check the paper's details, e.g., the math, experimental design, or novelty.

**Typos Grammar Style And Presentation Improvements:**

When introducing related works such as DPR, RAG, etc., you might want to provide the full name before using the abbreviation.

---

> ### Author Rebuttal · Authors · 2023-08-29
>
> We thank the reviewer for the valuable comments and feedback. We are glad that the reviewer finds our work investigating an important topic and conducting a wide range of empirical studies. We provide detailed responses below.
>
>
> > 1. “Too few baseline models are included in text-generation (section 5) and the LM evaluation harness benchmark (section6). Since the vanilla GPT is not aided by retrievals, it is not surprising to see it performs worse on knowledge-intensive tasks. The authors said "We observe that directly prepending the evidence from the retrieval database messes up the GPT context in the zero-shot setting", but a straightforward alternative is to write an instruction with few-shot examples. I would suggest the authors to include a few retrieval-based baselines, too, such as GPT + retrieval at inference time.”
>
>   - Thank you so much for your constructive suggestion. We'd also like to clarify that both Sections 5 (text generation) and Section 6 ([LM Evaluation Harness benchmark](https://github.com/EleutherAI/lm-evaluation-harness)) mainly focus on auto-regressive decoder models. Therefore, we mainly use GPT pretrained on the same 330B pretraining corpus as Retro as our primary baseline.
>
>   - Following your suggestion, we also add an additional baseline to the LM Evaluation Harness benchmark, which leverages retrieved information in the context of GPT models (GPT + In-context Retrieval). To better incorporate the retrieved evidence, we follow [1] and carefully design four different prompts and perform the zero-shot evaluation on the Hellaswag dataset for Small, Medium, and XL GPT and Retro. The results are shown below. Our results demonstrate that Retro can better leverage the retrieved evidence through the Retro encoder than GPT with in-context retrieval, because it is pretrained with retrieval.
>
> | Hellaswag |     GPT w/o retrieval       |GPT +  In-context  Retrieval (Template 1) | GPT +  In-context  Retrieval (Template 2) | GPT + In-context  Retrieval (Template 3) | GPT + In-context  Retrieval (Template 4) | Retro |
> | ---------|  ---------------------------- | ---------------------------- | ---------------------------- | ---------------------------- | ---------------------------- | -------- |
> | Small    | 31.3   | 34.4                        | 34.4                         | 34.5                        | 34.4                       |  **36.2**  |
> | Medium  | 43.2 | 44.0                           | 44.0                        | 44.1                        | 44.1                        |  **46.2**   |
> | XL       | 56.7    | 57.1                        | 57.3                        | 57.3                      | 57.5                        |  **59.0**    |
>
>
>
> > 2. “In Table 5 are the differences statistically significant enough?”
>
>   - Thanks for your comment. We note that our toxicity evaluation is run over 10k different prompts, and for each prompt, we generate 25 different generations with nucleus sampling to evaluate the worst-case toxicity. So the drop in the average toxicity probability from 40% to 35% is indeed a significant difference by just updating the retrieval database from the pretraining database to the Wiki database. Moreover, we observe a larger (6%) toxicity probability drop from 77% to 71% for toxic prompts, which further demonstrates the flexibility and potential of pretrained retrieval-augmented LLMs.
>
> > 3. “When introducing related works such as DPR, RAG, etc., you might want to provide the full name before using the abbreviation.”
>
> - Thank you so much for your valuable suggestions. DPR represents Dense Passage Retrieval [2], and RAG represents Retrieval-Augmented Generation [3]. We will follow your suggestion and add the full name before using the abbreviation in the revision.
>
>
> [1] Wei, Jason, Maarten Bosma, Vincent Zhao, Kelvin Guu, Adams Wei Yu, Brian Lester, Nan Du, Andrew M. Dai and Quoc V. Le. “Finetuned Language Models Are Zero-Shot Learners.” ArXiv abs/2109.01652 (2021): n. pag.
>
> [2] Vladimir Karpukhin, Barlas O˘guz, Sewon Min, Patrick Lewis, Ledell Wu, Sergey Edunov, Danqi Chen, and Wen-tau Yih. 2020. Dense passage retrieval for open-domain question answering. In EMNLP.
>
> [3] Patrick Lewis, Ethan Perez, Aleksandra Piktus, Fabio Petroni, Vladimir Karpukhin, Naman Goyal, Heinrich Küttler, Mike Lewis, Wen-tau Yih, Tim Rocktäschel, et al. 2020b. Retrieval-augmented generation for knowledge-intensive NLP tasks. In NeurIPS.

---

### Official Review · Reviewer_Q4QN · 2023-08-12

**Soundness:** 4

**Excitement:**

4: Strong: This paper deepens the understanding of some phenomenon or lowers the barriers to an existing research direction.

**Paper Topic And Main Contributions:**

The paper investigates the potential of pretraining large language models with a retrieval model. First the paper reproduces RETRO model and quantitatively and qualtiatively compares it with GPT. Furthermore, it introduces a new variant of RETRO model called RETRO++. The paper clearly demonstrates the performance gains of introducing a retrieval module in the pre-training stage and presents the limitations such as scalability and the impact of the retrieval database quality

**Questions For The Authors:**

What are the computation costs associated with training the retro and retro++ models? For the metrics generated, is it the average? what are the number of runs per experiment?

**Reasons To Accept:**

> The paper, in detail with very clear flow and writing, reproduces and builds upon previously published work to make the case for introducing a retrieval module in the pre-training stage.
> The paper clearly demonstrates useful contribution and research findings through the detailed discussion and quantitive and qualtative comparison between retrieval-based model and GPT

**Reasons To Reject:**

> The paper could use more models other than GPT for benchmarking
> The computation costs are not clearly mentioned
> For the metrics mentioned in the figures, is it average? the number of runs are not clearly mentioned?

**Reproducibility:**

5: Could easily reproduce the results.

**Reviewer Confidence:**

3: Pretty sure, but there's a chance I missed something. Although I have a good feel for this area in general, I did not carefully check the paper's details, e.g., the math, experimental design, or novelty.

---

> ### Author Rebuttal · Authors · 2023-08-29
>
> We thank the reviewer for the valuable comments and feedback. We are glad that the reviewer finds our work demonstrating useful contributions and research findings through detailed discussion and evaluations. We provide detailed responses below.
>
> > 1. “The paper could use more models other than GPT for benchmarking”
>
>   - Thank you for the valuable comments. In Section 7, we conduct an evaluation on two open-domain QA benchmarks and expand our comparisons beyond just GPT. Specifically, we incorporate encoder-decoder models, including RAG_{BART} and FiD_{Large}. The results illustrate that our Retro++ showcases significant improvements over the original Retro and other state-of-the-art models.
>
>   - We'd also like to clarify that both Sections 5 (text generation) and Section 6 ([LM Evaluation Harness benchmark](https://github.com/EleutherAI/lm-evaluation-harness)) mainly focus on auto-regressive decoder models. Therefore, we mainly use GPT pretrained on the same 330B pretraining corpus as Retro as our baseline.
>
>   - In addition, we also add an additional baseline to the LM Evaluation Harness benchmark, which leverages retrieved information in the context of GPT models (GPT + In-context Retrieval). To better incorporate the retrieved evidence, we follow [1] and carefully design four different prompts and perform the zero-shot evaluation on the Hellaswag dataset for Small, Medium, and XL GPT and Retro. The results are shown below. Our results demonstrate that Retro can better leverage the retrieved evidence through the Retro encoder than GPT with in-context retrieval, because it is pretrained with retrieval.
>
> | Hellaswag |     GPT w/o retrieval       |GPT +  In-context  Retrieval (Template 1) | GPT +  In-context  Retrieval (Template 2) | GPT + In-context  Retrieval (Template 3) | GPT + In-context  Retrieval (Template 4) | Retro |
> | ---------|  ---------------------------- | ---------------------------- | ---------------------------- | ---------------------------- | ---------------------------- | -------- |
> | Small    | 31.3   | 34.4                        | 34.4                         | 34.5                        | 34.4                       |  **36.2**  |
> | Medium  | 43.2 | 44.0                           | 44.0                        | 44.1                        | 44.1                        |  **46.2**   |
> | XL       | 56.7    | 57.1                        | 57.3                        | 57.3                      | 57.5                        |  **59.0**    |
>
>
> > 2. “The computation costs are not clearly mentioned. / What are the computation costs associated with training the retro and retro++ models?”
>
>   - Thank you for the valuable suggestion. The training cost of Retro++ is very close to Retro as Retro++ simply adds more context to the decoder at the fine-tuning stage. We have provided our computation costs associated with GPT and Retro below for pretraining on 330B tokens (All of our experiments are done on the DGX-2H node with 8x A100 GPUs).
>
> |        | GPT             | Retro           | Additional Overhead |
> | ------ | --------------- | --------------- | ------------------- |
> | Small  | 1240 GPU Hours  | 1560 GPU Hours  | 25.80%              |
> | Medium | 3600 GPU Hours  | 4480 GPU Hours  | 24.44%              |
> | XL     | 12000 GPU Hours | 13440 GPU Hours | 12.00%              |
>
>
>   - From the table, we can see that the overhead involved in training Retro is less than 25% on average. Considering consistent improvements in text generation quality, factual accuracy, lower toxicity, and downstream task accuracy, especially for knowledge-intensive tasks, including open-domain QA, we believe pretraining Retro is a promising direction.
>
>
> > 3. “For the metrics mentioned in the figures, is it average? the number of runs are not clearly mentioned?  / For the metrics generated, is it the average? what are the number of runs per experiment?”
>
>   - For the EM (Exact Match) scores mentioned in Figure 2, we first conduct grid searches to determine the best hyper-parameters (e.g., batch size from {256, 512, 768}  and learning rate from {5e-7, 1e-6, 3e-6, 5e-6, 1e-5, 5e-5}) for finetuning models of different sizes. Following the standard evaluation practice [2,3], we early-stop finetuning by evaluating the EM on the validation set and report the corresponding EM on the test set in Figure 2 and Table 7.
>
>   - We also follow your suggestion and repeat the fine-tuning experiments on Retro (Medium) on the NQ dataset for three runs with the best hyper-parameters and different random seeds. The EM scores for three runs are 0.4355, 0.4449, and 0.4521. The average EM score and the standard deviation are 0.4442 +- 0.0083, which is close to the numbers shown in Figure 2 (0.4449) and significantly better than RAG_{GPT} of 0.3958.
>
>
>
> [1] Wei, Jason, Maarten Bosma, Vincent Zhao, Kelvin Guu, Adams Wei Yu, Brian Lester, Nan Du, Andrew M. Dai and Quoc V. Le. “Finetuned Language Models Are Zero-Shot Learners.” ArXiv abs/2109.01652 (2021): n. pag.
>
> [2] Gautier Izacard and Édouard Grave. 2021. Leveraging passage retrieval with generative models for open domain question answering. In Proceedings of the 16th Conference of the European Chapter of the Association for Computational Linguistics: Main Volume, pages 874–880.
>
> [3] Patrick Lewis, Ethan Perez, Aleksandra Piktus, Fabio Petroni, Vladimir Karpukhin, Naman Goyal, Heinrich Küttler, Mike Lewis, Wen-tau Yih, Tim Rocktäschel, et al. 2020b. Retrieval-augmented generation for knowledge-intensive NLP tasks. In NeurIPS.

---

### Meta-Review · Area_Chair_6ZFC · 2023-09-25

**Recommendation:** 5

**Metareview:**

The paper explores the impact of retrieval on pretraining autoregressive LLMs. It reproduces the RETRO model and compares it with GPT, and proposes a new variant of it called RETRO++. The paper presents several useful experimental analyses and discussions such as performance gains and impact of retrieval database quality. The community can benefit from this work in furthering research on retrieval-augmented generation, which is becoming increasing important given the advent of LLMs.

For the authors — please include the additional experiments and insights you shared in the author responses, they would make the paper much stronger.

---

### Decision · Program_Chairs · 2023-10-07

**Decision:**

Accept-Main

**Comment:**

The paper explores the impact of retrieval on pretraining autoregressive LLMs. It reproduces the RETRO model and compares it with GPT, and proposes a new variant of it called RETRO++. The paper presents several useful experimental analyses and discussions such as performance gains and impact of retrieval database quality. The community can benefit from this work in furthering research on retrieval-augmented generation, which is becoming increasing important given the advent of LLMs.

For the authors — please include the additional experiments and insights you shared in the author responses, they would make the paper much stronger.